# Revisiting De-Identification of Electronic Medical Records: Evaluation of Within- and Cross-Hospital Generalization

**Yiyang Liu** and **Jinpeng Li** and **Enwei Zhu**[*]
Ningbo No.2 Hospital
Ningbo Institute of Life and Health Industry, University of Chinese Academy of Sciences
{liuyiyang,lijinpeng}@ucas.ac.cn, enwei.zhu@outlook.com

## Abstract

The de-identification task aims to detect and remove the protected health information from electronic medical records (EMRs). Previous studies generally focus on the *within-hospital* setting and achieve great successes, while the *cross-hospital* setting has been overlooked. This study introduces a new de-identification dataset comprising EMRs from three hospitals in China, creating a benchmark for evaluating both within- and cross-hospital generalization. We find significant domain discrepancy between hospitals. A model with almost perfect within-hospital performance struggles when transferred across hospitals. Further experiments show that pretrained language models and some domain generalization methods can alleviate this problem. We believe that our data and findings will encourage investigations on the generalization of medical NLP models.[1]

## 1 Introduction

De-identification is a natural language processing (NLP) task to detect and remove the protected health information (PHI) from electronic medical records (EMRs). It is a prerequisite to the distribution of EMRs outside their original institutions for medical NLP research (Uzuner et al., 2007).

Previous studies generally focus on the *within-hospital setting*, where the training and test data are from a same hospital or institution. This includes English tasks like 2006 i2b2 (Uzuner et al., 2007), 2014 i2b2/UTHealth (Stubbs et al., 2015a), 2016 CEGS N-GRID (Stubbs et al., 2017), and others in Swedish (Dalianis and Velupillai, 2010) and French (Grouin and Névéol, 2014). De-identification may be regarded as an easy task relative to other NLP tasks, because simple rule-based or shallow neural models can achieve 95%+ F1 scores (Liu et al., 2017; Dernoncourt et al., 2017).

However, the *cross-hospital setting* has been largely overlooked. This setting corresponds to a realistic scenario that a de-identification model is deployed across hospitals. This study aims to formally fill the gap. We introduce a new de-identification dataset comprising EMRs from three hospitals in China, establishing a benchmark for evaluating both within- and cross-hospital generalization. The latter poses a challenging domain generalization (DG) task, where hospitals are referred as domains.

We find significant domain discrepancy existing between hospitals. A model with almost perfect within-hospital performance encounters dramatic degradation when transferred to other hospitals. Using pretrained language models (PLMs) like BERT (Devlin et al., 2019), or some existing DG methods (Izmailov et al., 2018; Wu et al., 2022) can improve the cross-hospital generalization, but by limited margins. This paper contributes in twofold:

- To the best of our knowledge, our dataset provides the first de-identification benchmark that has multiple sources for cross-hospital evaluation. It is also the first de-identification task for Chinese EMRs. The dataset probably interests researchers from a broader community, since DG tasks have been scarce in NLP (Zhou et al., 2022). We will release the data to facilitate further research.

- From the DG perspective, our findings enhance Stubbs et al. (2015a, 2017)'s argument that de-identification is not a solved problem, even in the post-BERT era. Our experiments show the effectiveness of PLMs and DG methods, providing a promising direction for investigations on the cross-hospital generalization of medical NLP models.

---

[*]Corresponding author.
[1]Our data and code are available at https://github.com/lanyangyang93/Revisiting-De-Identification.

## 2 Data and Annotations

### 2.1 Data Sources

**HM.** Our primary dataset is built from the EMRs of HM[2] hospital, a general hospital in Zhejiang Province, China. We obtained a 1.7 TB backup of 671.5K inpatient records. The clinical text is stored in sections like chief complaints, examination reports, progress notes, and discharge summaries.

We sample 500 EMRs from 30 representative medical departments for PHI annotation. The resulting annotated corpus is randomly split into training/development/test sets with 300/100/100 EMRs. This is the dataset for within-hospital evaluation.

In addition, the HM database provides the large-scale clinical text for pretraining Word2Vec embeddings and BERT, named HM-Word2Vec and HM-BERT, respectively. This mitigates the potential domain gap of transferring word embeddings or PLMs trained on common corpora to clinical text. See Section 3 for more details.

**SY.** We collect 100 EMRs from SY[3] hospital, a general hospital in Hunan Province, China. Similar to those of HM hospital, the EMRs contain various sections of clinical text. After the annotation of PHI, SY dataset serves as a test set for cross-hospital evaluation.

**CCKS.** China Conference on Knowledge Graph and Semantic Computing (CCKS) 2017[4] released a clinical named entity recognition task, which contains 300 EMRs from an anonymous hospital in Hebei Province, China. Each EMR includes four paragraphs from specific sections. We annotate the PHI on the available text, yielding another test set.

### 2.2 Protected Health Information

Following previous research (Uzuner et al., 2007; Stubbs et al., 2017), we define eight PHI categories:

- PERSON: The names of patients or physicians.

- LOCATION: Addresses of patients.

- HOSPITAL: The names of hospitals.

- DATE: Date or time stamps.

- ID: IDs of patients or medical tests.

- CONTACT: The contact information of patients, physicians or hospitals.

---

[2]Ningbo No.2 Hospital.

[3]The hospital name is anonymized for policies and regulations.

[4]See https://www.sigkg.cn/ccks2017.

| | HM | | | SY | CCKS |
|---|---|---|---|---|---|
| | Train | Dev | Test | | |
| #EMR | 300 | 100 | 100 | 100 | 300 |
| #Sentence | 6,005 | 2,029 | 1,995 | 2,505 | 1,346 |
| #PHI Mention | | | | | |
| PERSON | 2,430 | 873 | 841 | 1,923 | 85 |
| LOCATION | 224 | 62 | 69 | 187 | 219 |
| HOSPITAL | 1,092 | 389 | 313 | 233 | 115 |
| DATE | 4,762 | 1,458 | 1,439 | 2,153 | 449 |
| ID | 129 | 32 | 31 | 22 | 9 |
| CONTACT | 207 | 64 | 67 | 36 | 0 |
| AGE | 1,087 | 401 | 370 | 253 | 430 |
| PROFESSION | 197 | 66 | 66 | 15 | 21 |

Table 1: Descriptive statistics of datasets.

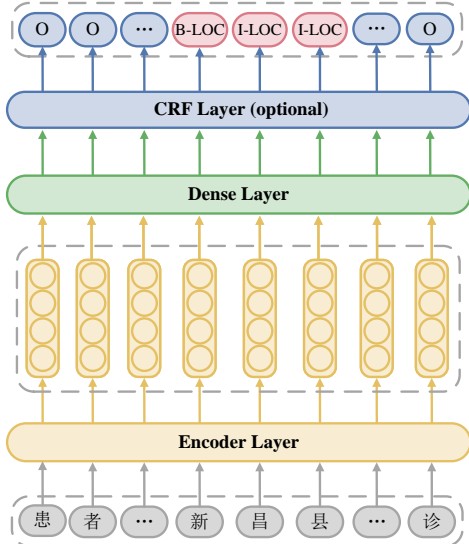

Figure 1: The architecture of our model.

- AGE: Ages of patients.

- PROFESSION: The professions of patients.

The BRAT Rapid Annotation Tool (Stenetorp et al., 2012) is employed for the PHI annotation. Table 1 reports the descriptive statistics of the resulting datasets. Appendix A provides some typical examples of the PHI annotations on clinical text.

## 3 Models and Experimental Settings

**Model Architecture.** Figure 1 displays the architecture of our model. We employ the neural sequence tagging framework (Collobert et al., 2011), a widely-used and mature solution for the de-identification task.

Specifically, the tokens are first mapped to embeddings and fed into the encoder. The encoder can be a 1D CNN (Zhang et al., 2015), a BiL-STM (Hochreiter and Schmidhuber, 1997b), or a pretrained Transformer (Devlin et al., 2019), which

transforms the embeddings to hidden representations. Finally, the dense layer classifies the representations into the pre-defined BIO tag space, and the resulting BIO tags can be parsed to identify the boundaries and categories of PHI mentions. An optional conditional random field (CRF) layer can be inserted after the dense layer, which may improve the consistency of the predicted BIO tags.

**Pretraining on EMRs.** Most publicly available Chinese word embeddings or PLMs are trained on common corpora, which may encounter domain gaps and result in sub-optimal performance when transferred to the clinical domain. To alleviate this issue, we pretrain Word2Vec embeddings (Mikolov et al., 2013) and a Chinese BERT (Devlin et al., 2019) on the large-scale clinical text from the HM database. The resulting embeddings and model are named HM-Word2Vec and HM-BERT, respectively.

After data parsing, cleaning and deduplication, the remaining HM corpus consists of 21.4K EMRs, including clinical text of 2.8 GB (1.1B tokens). HM-Word2Vec has a character-level vocabulary of size 5.7K and embedding size of 100. It is trained for 5 epochs by the Gensim (Řehůřek and Sojka, 2010) package with window size 15 and learning rate 1e-3.

Following Cui et al. (2021), we initialize HM-BERT from the *bert-base-chinese* checkpoint released by Hugging Face.[5] We then pretrain the model with the optimizer AdamW (Loshchilov and Hutter, 2018), learning rate 1e-4 and batch size 384 for 20 epochs on the whole corpus. A scheduler of linear warmup in the first 20% steps followed by linear decay is applied. The masking rate is 15% for the masked language modeling (MLM) task; the maximum input length is 512. In addition, we apply the whole word masking (Cui et al., 2021) and dynamic masking (Liu et al., 2019) strategies.

**Within- and Cross-Hospital Evaluation.** We train the models on the training set of HM, and use the development set for hyperparameter tuning. We perform the *within-hospital evaluation*, i.e., evaluating the trained models on the test set of HM; and the *cross-hospital evaluation*, i.e., evaluating the models on SY and CCKS.

[5]See https://huggingface.co/bert-base-chinese.

**Hyperparameters.**[6] For CNN or BiLSTM models, the embedding layer is 100-dimensional, and optionally initialized from HM-Word2Vec; the encoders have one layer with 200 hidden states and dropout rate of 0.5. The kernel size is 15 for CNN. The models are trained for 100 epochs by the AdamW (Loshchilov and Hutter, 2018) optimizer with learning rate 1e-3 and batch size 32.

For models with PLMs, we use BERT-wwm (Cui et al., 2021), MC-BERT (Zhang et al., 2020) and HM-BERT, all of a base size (768 hidden size, 12 layers). The MC-BERT is pre-trained on Chinese medical corpora including biomedical question answering, medical encyclopedia and EMRs. The models are trained for 100 epochs by the AdamW optimizer with learning rate 2e-5 and batch size 32.

**Evaluation.** A predicted PHI mention is considered correct if its boundaries and category exactly match the ground truth. The evaluation metrics are micro precision rate, recall rate and F1 score on the test sets. All the experiments are repeated for five times and the average metrics are reported.

## 4 Experimental Results

### 4.1 Main Results

Table 2 presents the results for both within- and cross-hospital evaluation. In the within-hospital evaluation, a single-layer CNN or BiLSTM can achieve 98% F1 scores or higher. This is consistent with the results of previous literature that the de-identification task can be almost perfectly solved by simple models (Dernoncourt et al., 2017; Liu et al., 2017). We add to the findings that more sophisticated neural models like BERT can further improve the performance, although by limited magnitudes.

However, the cross-hospital setting has largely been overlooked in literature. With the help of our multi-source data, we evidence that a decent neural de-identification model easily encounters noticeable performance degradation when transferred across hospitals. Specifically, for CNN and BiLSTM, the F1 scores decrease to 70%–80% and 50%–60% when transferred to SY and CCKS, respectively. The HM-Word2Vec embeddings and CRF help to resist the performance drop, but the effect is not robust. The BERT-based models also suffer from the cross-hospital setting, but they achieve much better results: 95%+ F1 scores on SY and

[6]All the hyperparameters have been extensively tuned and thus empirically optimal. We have tested models with larger sizes, but resulted in lower performance.

| | HM→HM | | | HM→SY | | | HM→CCKS | | | #Params | FLOPs | Speed (sents/s) |
|---|---|---|---|---|---|---|---|---|---|---|---|---|
| | Prec. | Rec. | F1 | Prec. | Rec. | F1 | Prec. | Rec. | F1 | | | |
| CNN | 97.9 | 98.6 | 98.2 | 70.6 | 76.4 | 73.4 | 51.3 | 57.6 | 54.2 | 576.6K | 101.3M | 500 |
| + HM-Word2Vec | 97.5 | 98.5 | 98.0 | 76.6 | 80.9 | 78.7 | 49.7 | 55.0 | 52.2 | 576.6K | 101.3M | 500 |
| + CRF | 95.4 | 95.0 | 95.2 | 70.1 | 66.8 | 68.4 | 64.8 | 59.3 | **61.9** | 577.0K | 101.3M | 25 |
| BiLSTM | 98.4 | 98.9 | 98.6 | 80.2 | 77.7 | 78.9 | 48.5 | 58.2 | 52.8 | 437.6K | 51.0M | 272 |
| + HM-Word2Vec | 98.8 | 99.0 | **98.9** | 82.6 | 82.7 | **82.6** | 44.8 | 55.0 | 49.3 | 437.6K | 51.0M | 272 |
| + CRF | 97.6 | 96.8 | 97.2 | 78.5 | 69.3 | 73.6 | 64.2 | 58.4 | 61.0 | 438.0K | 51.0M | 25 |
| BERT-wwm | 99.3 | 99.5 | 99.4 | 96.2 | 97.8 | 97.0 | 75.9 | 77.5 | 76.7 | 102.6M | 26.4G | 22 |
| + BiLSTM | 99.5 | 99.4 | 99.5 | 97.0 | 97.8 | **97.4** | 74.4 | 72.9 | 73.7 | 103.3M | 26.6G | 21 |
| + CRF | 99.1 | 99.2 | 99.1 | 94.9 | 96.7 | 95.8 | 70.3 | 71.3 | 70.8 | 102.6M | 26.4G | 13 |
| MC-BERT | 99.4 | 99.5 | 99.5 | 96.2 | 97.8 | 97.0 | 75.9 | 77.5 | 76.7 | 102.6M | 26.4G | 22 |
| + BiLSTM | 99.5 | 99.5 | 99.5 | 96.5 | 97.8 | 97.2 | 74.7 | 77.7 | 76.1 | 103.3M | 26.6G | 21 |
| + CRF | 99.0 | 99.1 | 99.0 | 94.8 | 96.7 | 95.7 | 70.3 | 71.3 | 70.8 | 102.6M | 26.4G | 15 |
| HM-BERT | 99.6 | 99.8 | **99.7** | 96.2 | 97.2 | 96.7 | 73.6 | 84.0 | 78.5 | 102.6M | 26.4G | 22 |
| + BiLSTM | 99.7 | 99.7 | **99.7** | 96.6 | 97.8 | 97.2 | 78.2 | 85.7 | **81.8** | 103.3M | 26.6G | 21 |
| + CRF | 99.4 | 99.7 | 99.5 | 92.7 | 95.5 | 94.1 | 70.8 | 79.4 | 74.8 | 102.6M | 26.4G | 15 |

Table 2: Results of within- and cross-hospital evaluation of de-identification. The models are trained on the training set of HM, and evaluated on the test set of HM, SY and CCKS, respectively.

70%–80% on CCKS. In particular, HM-BERT outperforms BERT-wwm and MC-BERT on CCKS.

These results clearly reveal a noteworthy problem – severe domain discrepancy exists between EMRs from different hospitals. It may greatly impede the cross-hospital applicability of a perfectly-performing model. Empirically, using PLMs can effectively alleviate this problem. The PLMs learn universal linguistic patterns from large-scale pre-training data, which help the models to generalize across hospitals.

When focused on within-hospital evaluation, we may conclude that CNN or BiLSTM models are superior to BERT-based models, because the former ones achieve similar performance with better efficiency (fewer parameters and FLOPs, higher speed). However, with the awareness of cross-hospital results, we have to rethink this problem.

Appendix B reports the categorical evaluation results. To avoid PHI leaks and preserve the data usability, we have carefully replaced the PHI mentions by realistic surrogates (Stubbs et al., 2015b) in the release version. This slightly affects the evaluation results, so we report the corresponding results in Appendix D.

## 4.2 Analysis

We explore some potential reasons for the significant gap between the within- and cross-hospital performance.

**Cosine Similarity.** Following Elangovan et al. (2021), we represent each data instance with the vector of HM-BERT and compute the cosine sim-

| | ACS | FAR95 |
|---|---|---|
| HM-train vs. HM-test | 92.05 | 94.64 |
| HM-train vs. SY | 84.50 | 92.05 |
| HM-train vs. CCKS | 82.51 | 73.03 |

Table 3: ACS and FAR95 of HM training set against HM test set, SY and CCKS sets.

ilarities between them. The average cosine similarity (ACS) over all the test instances is used as an indicator to measure the overlapping extent of training and test data.

**False Alarm Rate.** Following Hendrycks et al. (2020), we assign the maximum softmax anomaly score for each test instance, to perform out-of-distribution (OOD) detection, and report the false alarm rate at 95% recall (FAR95).

Table 3 shows that the CCKS data present the most different distributions from the HM training set. In other words, the overlap between HM and CCKS is the lowest. Hence, the CCKS has representations of more OOD information that the models fail to learn. This is the major reason for the large performance drop on CCKS. We further perform the PCA visualizations of sentence representations on the HM training/test sets, SY, and CCKS. The results are quite similar. See Appendix C for more details.

## 4.3 Domain Generalization

Given the domain shift between hospitals, we explore some widely used DG methods, verifying whether they can help the models to generalize

| | CNN | | | HM-BERT | | |
|---|---|---|---|---|---|---|
| | HM→HM | HM→SY | HM→CCKS | HM→HM | HM→SY | HM→CCKS |
| Baseline (w/o DG methods) | 98.2 | 73.4 | 54.2 | 99.7 | 96.7 | 78.5 |
| Mention Substitution | 98.3 (+0.1) | 56.7 (-16.7) | 57.2 (+3.0) | 99.4 (-0.3) | 92.2 (-4.5) | 82.3 (+3.8) |
| Text Smoothing | – | – | – | 99.7 (+0.0) | 97.0 (+0.3) | 80.7 (+2.2) |
| Stochastic Weight Averaging | | | | | | |
| 25% | 98.3 (+0.1) | 74.7 (+1.3) | 54.3 (+0.1) | 99.8 (+0.1) | 97.2 (+0.5) | 80.5 (+2.0) |
| 50% | 98.3 (+0.1) | 74.5 (+1.1) | 55.5 (+1.3) | 99.8 (+0.1) | 97.3 (+0.6) | 79.4 (+0.9) |
| 75% | 98.3 (+0.1) | 74.3 (+0.9) | 56.1 (+1.9) | 99.8 (+0.1) | 97.4 (+0.7) | 80.6 (+2.1) |
| Dropout (def. = 0.50) | | | | | | |
| 0.25 | 98.4 (+0.2) | 75.5 (+0.2) | 50.1 (-4.1) | 99.6 (-0.1) | 96.7 (+0.0) | 77.7 (-0.8) |
| 0.75 | 94.8 (-3.4) | 66.4 (-7.0) | 48.7 (-5.5) | 99.7 (+0.0) | 96.7 (+0.0) | 81.1 (+2.6) |
| L2 Regularization (def. = 0.01) | | | | | | |
| 0.05 | 98.0 (-0.2) | 74.1 (+0.7) | 51.1 (-3.1) | 99.6 (-0.1) | 96.2 (-0.5) | 79.5 (+1.0) |
| 0.10 | 98.0 (-0.2) | 74.3 (+0.9) | 50.5 (-2.7) | 99.7 (+0.0) | 96.8 (+0.1) | 78.4 (-0.1) |

Table 4: Results of within- and cross-hospital evaluation of domain generalization methods. The models are trained on the training set of HM, and evaluated on the test set of HM, SY and CCKS, respectively.

across hospitals. The results are shown in Table 4.

**Mention Substitution.** Data augmentation has been a common practice to prevent the models from overfitting and thus improve generalization (LeCun et al., 2015; Zhou et al., 2022). To avoid overfitting to frequent PHI mentions, we add an augmentation module that randomly replaces the PHI mentions with fabricated ones of the same categories. For example, a LOCATION mention can be replaced by random addresses. The resulting effect is unstable.

**Text Smoothing.** Text smoothing (Wu et al., 2022) is a PLM-based text augmentation approach. It leverages the masked language modeling objective of a dedicated PLM and augments each token according to the predicted probabilities over the vocabulary. Text smoothing slightly improves the generalization performance on SY and CCKS.

**Stochastic Weight Averaging.** Stochastic weight averaging (Izmailov et al., 2018, SWA) aggregates the model weights along the training trajectory. The ensemble model can achieve flatter minima (Hochreiter and Schmidhuber, 1997a) and thus improves generalization. Specifically, we average the model checkpoints at last 25%, 50%, and 75% training epochs for test. It shows that SWA brings marginal yet robust improvements for either within- or cross-hospital performance.

**Dropout and L2 Regularization.** Dropout (Srivastava et al., 2014) and L2 regularization are standard strategies against overfitting. We tune the dropout rate and L2 penalty to different values, but obtain lower scores.

In summary, text smoothing and SWA can effectively improve the generalization performance by small margins, while the other DG methods result in negative or unstable effects.

## 5 Conclusion

In this paper, we revisit the EMR de-identification task and create a new dataset. It consists of EMRs from three hospitals and thus asks evaluation for both within- and cross-hospital generalization. The latter poses a challenging DG task, which corresponds to a realistic scenario that a de-identification model is required to deploy across hospitals.

With this new benchmark, we find significant domain shift between hospitals. A model with perfect within-hospital performance struggles when transferred across hospitals. Using PLMs or some existing DG methods can alleviate but not address this problem.

## 6 Limitations

Although we have explored various existing models and DG methods on our proposed task, there are still other approaches worth investigating. It may be more effective to develop new specialized methods to improve the cross-hospital generalization of medical NLP models.

In general, this study focuses on end-to-end evaluation. It requires more in-depth analysis, either theoretically or empirically, to answer some crucial questions like (1) how and to what extent is the i.i.d. assumption violated between the clinical text from different hospitals? (2) how to develop invariant representations that generalize across hospitals? (3)

how to estimate the risk of generalization failure before the models are deployed? These answers may be the key towards interpretable, robust and reliable medical NLP systems.

## 7 Ethical Considerations

Our data were collected and used consistently with the terms of use. The EMRs of HM and SY were obtained from the authors' affiliations, and this work was performed as a part of projects approved by the ethics review committees. CCKS was derived from publicly available data released as a shared task. We authors performed the PHI annotation, with full awareness of the potential impacts and risks.

The data of release version have been de-identified. Specifically, we have carefully replaced all the PHI mentions by surrogates (Stubbs et al., 2015b), and manually verified the resulting text so that the risk of privacy leak has been minimized. In particular, any text span is regarded as PHI and removed if it potentially reveals any identity characteristics, even if the risk is impossibly low. In addition, the data will be released upon a data use agreement that forbids any inappropriate use, especially identification of any individuals or institutions.

We report data characteristics and experimental results averaged over multiple runs. We will release the data and code, ensuring the reproducibility of our work. Our experiments do not require high computational cost, relative to pretraining PLMs.

## Acknowledgements

We thank the anonymous reviewers for their insightful comments and feedback. This work is supported by Zhejiang Provincial Natural Science Foundation of China (No. LQ23F020005), National Natural Science Foundation of China (No. 62106248), Ningbo Science and Technology Service Industry Demonstration Project (No. 2020F041), and Ningbo Public Service Technology Foundation (No. 2021S152).

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

# A Examples of PHI Annotations

All the EMRs are manually annotated by two native speakers. (A master's degree in computer science and an expert in both the fields of computer science and medicine.) We consider the de-identification task to be easy and straightforward, with very few disagreements during labeling and near-perfect accuracy. We employed some post-processing procedure to ensure the accuracy of the annotations. For example, we perform cross-validation on the whole dataset, and manually check the inconsistencies between the predicted PHI mentions and ground-truth.

Table 5 shows examples of clinical text and PHI annotations. In principle, a text span is regarded as PHI if it potentially reveals any personal information, even if the risk is impossibly low.

Some text spans are excluded because of irrelevance to personal information, although they seemingly fall into specific PHI categories. For example, some diseases are named after persons or locations, and these person or location names should not be regarded as PHI. Some clinical text may describe general medical knowledge that relates to ages or jobs, which are also excluded in our annotation.

# B Categorical Results

Table 6 presents the evaluation results by PHI categories. The cross-hospital F1 scores are lower than the corresponding within-hospital scores across all categories, while such effect is heterogeneous. Taking HM-BERT as the example, it generalizes relatively well on PERSON, DATE and AGE categories,

| Category | Clinical text and PHI annotations |
|---|---|
| PERSON | [姜淑]副主任医师查房. |
| | [Shu Jiang], associate chief physician, took ward rounds. |
| | 患者七年前得了[帕金森]综合征. |
| | The patient got [Parkinson]'s syndrome seven years ago. |
| LOCATION | 患者出生于[新昌县]，原籍长大. |
| | The patient was born in [Xinchang County] and grew up there. |
| | [埃博拉]病毒会导致发烧、喉咙痛、肌肉痛和头痛. |
| | [Ebola] virus disease causes fever, sore throat, muscle pain, and headaches. |
| HOSPITAL | 患者至[上海中医院]就诊. |
| | The patient went to [Shanghai Traditional Chinese Medicine Hospital] for diagnosis. |
| DATE | [2019年10月2日]，患者因腹痛收治入院. |
| | [October 2, 2019], the patient was admitted because of abdominal pain. |
| ID | 胸部CT（[US00786]）提示气管狭窄. |
| | Chest CT ([US00786]) suggests tracheal stenosis. |
| CONTACT | 门诊预约电话：[88121834]. |
| | Outpatient phone number: [88121834]. |
| | 急救电话：[120]. |
| | Emergency phone number: [120]. |
| AGE | 患者（[68岁]）因踝关节骨折入院. |
| | The patient ([68 years old]) was admitted for ankle fracture. |
| | 高血压易出现在[80岁]以上人群中. |
| | High blood pressure tends to occur in people over [80 years old]. |
| PROFESSION | 患者，男，初中文化，职业：[工人]. |
| | The patient, male, graduated from a junior high school, was a [worker]. |
| | [矿工]和[土建工人]易患尘肺病. |
| | [Miners] and [construction workers] are vulnerable to pneumoconiosis. |

Table 5: Examples of clinical text and PHI annotations. The Chinese texts are translated to English for reference. The PHI mentions are marked in red [*], while the non-PHI but possibly confusing mentions are marked in blue [*].

| | CNN | | | HM-BERT | | |
|---|---|---|---|---|---|---|
| | HM→HM | HM→SY | HM→CCKS | HM→HM | HM→SY | HM→CCKS |
| PERSON | 98.7 | 69.2 | 48.9 | 99.8 | 98.0 | 93.2 |
| LOCATION | 94.2 | 22.0 | 6.2 | 97.6 | 92.9 | 30.7 |
| HOSPITAL | 95.9 | 43.8 | 25.0 | 99.2 | 95.3 | 66.7 |
| DATE | 98.3 | 86.1 | 76.3 | 99.8 | 97.0 | 89.0 |
| ID | 89.9 | 13.5 | 24.2 | 97.8 | 78.9 | 65.3 |
| CONTACT | 99.7 | 15.5 | – | 100.0 | 85.7 | – |
| AGE | 100.0 | 95.7 | 72.8 | 100.0 | 96.4 | 98.4 |
| PROFESSION | 99.2 | 31.2 | 11.5 | 99.2 | 45.5 | 40.3 |
| Overall | 98.2 | 73.4 | 54.2 | 99.7 | 96.7 | 78.5 |

Table 6: Categorical F1 scores of within- and cross-hospital evaluation. The models are trained on the training set of HM, and evaluated on the test set of HM, SY and CCKS, respectively.

but struggles on LOCATION, ID and PROFESSION. This plausibly attributes to that the former PHI categories are typically associated with clearer language patterns than the latter ones.

## C Visualization of Sentence Representations

Figure 2 displays the PCA visualizations of sentence representations on the HM training/test sets, SY, and CCKS. The sentence representations of the HM training and test sets are most similar and overlapping. However, there are significant differences in the distributions of the HM and CCKS datasets in the feature space. In comparison, there is a small overlap between the SY dataset and the HM dataset.

Further underlying reasons for the differences in the data distributions would be complicated. For example, different doctors may record information differently, resulting in differences in the format and content of medical records. Different medical institutions may use different electronic medical record systems.

| | HM→HM | | | HM→SY | | | HM→CCKS | | |
|---|---|---|---|---|---|---|---|---|---|
| | Prec. | Rec. | F1 | Prec. | Rec. | F1 | Prec. | Rec. | F1 |
| CNN | 97.8 | 98.5 | 98.1 | 73.4 | 75.0 | 74.2 | 48.5 | 55.3 | 51.7 |
| + HM-Word2Vec | 97.8 | 98.7 | 98.3 | 75.3 | 79.5 | 77.4 | 50.8 | 54.1 | 52.4 |
| + CRF | 96.1 | 95.5 | 95.8 | 75.0 | 70.5 | 72.7 | 63.1 | 58.9 | **60.9** |
| BiLSTM | 98.4 | 98.9 | 98.6 | 78.8 | 77.8 | 78.3 | 45.0 | 56.6 | 50.1 |
| + HM-Word2Vec | 98.8 | 99.0 | **98.9** | 80.7 | 81.4 | **81.0** | 49.2 | 57.4 | 52.9 |
| + CRF | 97.7 | 97.1 | 97.4 | 83.2 | 75.3 | 79.0 | 57.5 | 55.1 | 56.2 |
| BERT-wwm | 99.4 | 99.8 | 99.6 | 95.8 | 97.5 | 96.7 | 76.2 | 80.4 | 78.2 |
| + BiLSTM | 99.5 | 99.6 | 99.6 | 96.6 | 98.0 | **97.3** | 78.1 | 78.8 | 78.4 |
| + CRF | 99.1 | 99.5 | 99.3 | 94.5 | 97.1 | 95.8 | 75.4 | 75.3 | 75.3 |
| MC-BERT | 99.4 | 99.5 | 99.5 | 95.8 | 97.7 | 96.7 | 74.9 | 78.5 | 76.6 |
| + BiLSTM | 99.4 | 99.7 | 99.6 | 96.2 | 98.0 | 97.1 | 75.5 | 75.2 | 75.3 |
| + CRF | 99.2 | 99.5 | 99.4 | 94.1 | 96.8 | 95.4 | 76.5 | 75.2 | 75.8 |
| HM-BERT | 99.6 | 99.8 | **99.7** | 95.6 | 97.4 | 96.5 | 75.5 | 84.6 | 79.7 |
| + BiLSTM | 99.6 | 99.6 | 99.6 | 96.3 | 97.6 | 97.0 | 80.7 | 85.9 | **83.2** |
| + CRF | 99.4 | 99.6 | 99.5 | 93.1 | 95.9 | 94.5 | 73.4 | 78.2 | 75.7 |

Table 7: Results of within- and cross-hospital evaluation on data of release version. The models are trained on the training set of HM, and evaluated on the test set of HM, SY and CCKS, respectively.

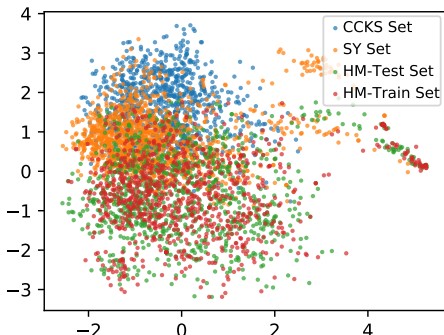

Figure 2: The PCA visualizations of sentence representations on the HM training/test sets, SY, and CCKS.

# D    Results on Data of Release Version

We cannot release our data with the real PHI mentions. Hence, we have carefully replaced the PHI mentions by realistic surrogates (Stubbs et al., 2015b). For example, the PERSON mentions are replaced by combinations of randomly sampled family and given names, where the sampling accords to the frequencies reported by National Bureau of Statistics of China. The LOCATION mentions are replaced by randomly sampled addresses in China. Such process preserves the usability of our data and prevent PHI leak simultaneously.

Table 7 presents the corresponding evaluation results, which are highly consistent with those on the original data. Hence, experimental results reported on our release data are still indicative.