# OpenReview forum: "Revisiting De-Identification of Electronic Medical Records: Evaluation of Within- and Cross-Hospital Generalization"
_EMNLP/2023/Conference — EMNLP 2023 Main_

### Official Review · Reviewer_iRFx · 2023-07-23

**Soundness:** 4

**Excitement:**

3: Ambivalent: It has merits (e.g., it reports state-of-the-art results, the idea is nice), but there are key weaknesses (e.g., it describes incremental work), and it can significantly benefit from another round of revision. However, I won't object to accepting it if my co-reviewers champion it.

**Missing References:**

1) Memorization vs. Generalization : Quantifying Data Leakage in NLP Performance Evaluation (Elangovan et al., EACL 2021)
2) Pretrained Transformers Improve Out-of-Distribution Robustness (Hendrycks et al., ACL 2020)

**Paper Topic And Main Contributions:**

This paper studies generalization gap cross hospitals for the task of PHI identification using electronic health records from 3 hospitals. This paper demonstrates that despite high test set performance for "within-hospital", there is upto 20 points drop in cross hospital performance even for a relatively simple task  of PHI identification of specific named entities (*location, name of the patient, hospital, patient id, date/timestamps, contact information of the patient/physician/hospital, profession/age of the patient)*.

**Questions For The Authors:**

- A) Are you able to include some analysis of what could possibly be some of the reasons that the performance gap is higher between hospitals "HM" and "CCKS"?  Some existing work that analyze  performance gap includes this paper for NER 1) *"Memorization vs. Generalization : Quantifying Data Leakage in NLP Performance Evaluation (Elangovan et al., EACL 2021)"* 2)*Pretrained Transformers Improve Out-of-Distribution Robustness (Hendrycks et al., ACL 2020)*

**Reasons To Accept:**

- The paper is well written and clear.
- The main strength of this paper is highlighting the generalization gap (upto 20 points from in cross hospital setting ) in deep learning models.

**Reasons To Reject:**

- A) While the paper highlights the generalization  gap ( an important topic), there is limited analysis of what could be causing the generalization gap? For instance, what are the similarities and dissimilarities between the data from these hospitals that might be causing model performance to drop by upto 20 points?

**Reproducibility:**

3: Could reproduce the results with some difficulty. The settings of parameters are underspecified or subjectively determined; the training/evaluation data are not widely available.

**Reviewer Confidence:**

3: Pretty sure, but there's a chance I missed something. Although I have a good feel for this area in general, I did not carefully check the paper's details, e.g., the math, experimental design, or novelty.

**Typos Grammar Style And Presentation Improvements:**

- A) Please move some of the pretraining details (use of HM-BERT is based on Bert-base-chinese, and how HM-Word2Vec is created ) from the appendix to the main section. This is important for the readers to understand the context of how the custom  pretrained models / word2vec embeddings were created and how they contribute to generalization gap

- B) Can you  briefly explain "BERT-wwm" in line 157? and how it is different from BERT-base-chinese?

---

> ### Author Rebuttal · Authors · 2023-08-29
>
> Thank you for the constructive and insightful comments. We reply to your comments as follows:
>
> > A) While the paper highlights the generalization gap ( an important topic), there is limited analysis of what could be causing the generalization gap? For instance, what are the similarities and dissimilarities between the data from these hospitals that might be causing model performance to drop by upto 20 points?
>
> > A) Are you able to include some analysis of what could possibly be some of the reasons that the performance gap is higher between hospitals "HM" and "CCKS"? Some existing work that analyze performance gap includes this paper for NER 1) "Memorization vs. Generalization : Quantifying Data Leakage in NLP Performance Evaluation (Elangovan et al., EACL 2021)" 2)Pretrained Transformers Improve Out-of-Distribution Robustness (Hendrycks et al., ACL 2020)
>
> * Thanks for your suggestion. The below table reports the results:
>     * **Cosine Similarity**: Following Elangovan et al. (2021), we represent each train/test instance with a vector of HM-BERT and compute the cosine similarities. We then use the average similarity over all the test instances as an indicator to measure the extent of train/test overlap.
>     * **False Alarm Rate**: Following Hendrycks et al. (2020), we assign each test instance the maximum softmax anomaly score, to perform out-of-distribution (OOD) detection,  and report the False Alarm Rate at 95% Recall (FAR95).
> * The results show that the CCKS data present the most different distributions from HM train set; in other words, the overlap between HM and CCKS is the lowest. Hence, the CCKS has representations of out-of-distribution information that models fail to learn. This is the major reason for the large performance drop on CCKS.
>
> |   |Elangovan et al., EACL 2021 (cosine similarity)  |Hendrycks et al., ACL 2020 (False Alarm Rate at 95% Recall)  |
> |:--|:--:|:--:|
> | HM train vs. HM test | 92.05%|  94.64% |
> | HM train vs. SY |  84.50%| 92.05% |
> | HM train vs. CCKS |  82.51% |  73.03%|
>
> * Further underlying reasons for differences in the data distributions would be complicated. For example, different doctors may record information differently, resulting in differences in the format and content of medical records. Different medical institutions may use different electronic medical record systems.
> * We will add the above analysis in our revised manuscript.
>
>
> > Missing References:
> > Memorization vs. Generalization : Quantifying Data Leakage in NLP Performance Evaluation (Elangovan et al., EACL 2021)
> > Pretrained Transformers Improve Out-of-Distribution Robustness (Hendrycks et al., ACL 2020)
> * We will cite the papers together with the analysis on performance gap.
>
>
> > A) Please move some of the pretraining details (use of HM-BERT is based on Bert-base-chinese, and how HM-Word2Vec is created ) from the appendix to the main section. This is important for the readers to understand the context of how the custom pretrained models / word2vec embeddings were created and how they contribute to the generalization gap
> * We sincerely appreciate your suggestions. We will revise the paper accordingly.
>
> > B) Can you briefly explain "BERT-wwm" in line 157? and how it is different from BERT-base-chinese?
> * BERT-base-Chinese is the original Chinese BERT released by Google.
> * BERT-wwm (Cui et al., 2021) stands for "BERT with whole word masking", which is initialized from BERT-base-Chinese, and further pre-trained on a large Chinese corpus. During the pre-training, the whole word masking strategy is applied in the masked language modeling to enhance the model performance. BERT-wwm outperforms BERT-base-Chinese on many Chinese NLP tasks.
> * Our HM-BERT is also initialized from BERT-base-Chinese, but pre-trained on HM database. According to our experiments, HM-BERT outperforms BERT-base-Chinese and BERT-wwm on Chinese medical NLP tasks.
>
> 1. Cui et al. 2021. Pre-training with whole word masking for Chinese BERT. *IEEE/ACM Transactions on Audio, Speech, and Language Processing*, 29: 3504-3514.

---

### Official Review · Reviewer_HyJ9 · 2023-08-02

**Soundness:** 3

**Excitement:**

4: Strong: This paper deepens the understanding of some phenomenon or lowers the barriers to an existing research direction.

**Paper Topic And Main Contributions:**

This paper studies de-identification of clinical notes, focusing on a cross-hospital setting. The main contribution is a manually PHI-annotated corpus of clinical notes from three Chinese hospitals. The authors also evaluate fine-tuned Chinese BERT-based models on this corpus, and compare same-hospital and cross-hospital settings. They show that performance can vary quite a lot across hospitals and that the de-identification task is not yet solved by current approaches.

**Questions For The Authors:**

A) What are the specificities of the CCKS dataset? Do the authors have any hypothesis regarding the drop in performance? Could they identify any major differences between the documents?

B) Details on the annotation procedure: How many annotators were there? What was their expertise? Was it a single or a double annotation (i.e. was each report annotated by one annotator, or by two annotators followed by adjudication over disagreements?)? Was inter-annotator agreement computed?

C) What are the descriptive statistics of each data split in the HM corpus?



**Reasons To Accept:**

- The paper tackles an important issue for medical NLP applications
- The dataset is very valuable, as clinical corpora are not numerous, particularly cross-hospital and non-English ones.


**Reasons To Reject:**

- The authors chose to build their own medical BERT-based model (HM-BERT) and ignored existing pre-trained medical models such as that of Zhang et al. 2020 (see ref below). It would have been interesting to test both models.
- The analysis of the results could have been more thorough. Specifically, investigating the reason for the much lower performance on the CCKS dataset would have been interesting (see questions below).

Ref:
Ningyu Zhang, Qianghuai Jia, Kangping Yin, Liang Dong, Feng Gao, and Nengwei Hua. 2020. Conceptualized representation learning for Chinese biomedical text mining. arXiv preprint arXiv:2008.10813.

**Reproducibility:**

3: Could reproduce the results with some difficulty. The settings of parameters are underspecified or subjectively determined; the training/evaluation data are not widely available.

**Reviewer Confidence:**

3: Pretty sure, but there's a chance I missed something. Although I have a good feel for this area in general, I did not carefully check the paper's details, e.g., the math, experimental design, or novelty.

**Typos Grammar Style And Presentation Improvements:**

page 1, line 35: "may be regarded as a easy task" -> "may be regarded as an easy task"

---

> ### Author Rebuttal · Authors · 2023-08-29
>
> We greatly appreciate your insightful and constructive comments. We reply to your concerns as follows:
>
> > The authors chose to build their own medical BERT-based model (HM-BERT) and ignored existing pre-trained medical models such as that of Zhang et al. 2020 (see ref below). It would have been interesting to test both models.
> * We greatly appreciate your suggestion. The below table shows our evaluation results of MC-BERT vs. HM-BERT. The results are quite similar. The de-identification model encounters significant performance degradation when transferred across hospitals.
>
> |   |MC-BERT  | HM-BERT  |
> |--|--|--|
> | HM->HM   | 99.4 | 99.7 |
> | HM->SY   | 97.0 | 96.7 |
> | HM->CCKS | 76.7 | 78.5 |
>
> > The analysis of the results could have been more thorough. Specifically, investigating the reason for the much lower performance on the CCKS dataset would have been interesting (see questions below).
>
> > A) What are the specificities of the CCKS dataset? Do the authors have any hypothesis regarding the drop in performance? Could they identify any major differences between the documents?
> * Many thanks for the question. Following another reviewer's suggestion, we calculate the cosine similarities and perform out-of-distribution (OOD) detection between the HM training set and the three test sets. The below table reports the results. The results show that the CCKS data present the most different distributions from HM train set; in other words, the overlap between HM and CCKS is the lowest. Hence, the CCKS has representations of out-of-distribution information that models fail to learn. This is the major reason for the large performance drop on CCKS.
>
> |   |Elangovan et al., EACL 2021 (cosine similarity)  |Hendrycks et al., ACL 2020 (False Alarm Rate at 95% Recall)  |
> |:--|:--:|:--:|
> | HM train vs. HM test | 92.05%|  94.64% |
> | HM train vs. SY |  84.50%| 92.05% |
> | HM train vs. CCKS |  82.51% |  73.03%|
>
> * We further perform the PCA visualizations of sentence representations on the HM training/test sets, SY, and CCKS. (It seems that we cannot upload the figure to OpenReview.) The sentence representations of the HM training and test sets are most similar and overlapping. However, there are significant differences in the distributions of the HM and CCKS datasets in the feature space. In comparison, there is a small overlap between the SY dataset and the HM dataset.
> * Further underlying reasons for differences in the data distributions would be complicated. For example, different doctors may record information differently, resulting in differences in the format and content of medical records. Different medical institutions may use different electronic medical record systems.
> * We will add the above analysis in our revised manuscript.
>
>
> > B) Details on the annotation procedure: How many annotators were there? What was their expertise? Was it a single or a double annotation (i.e. was each report annotated by one annotator, or by two annotators followed by adjudication over disagreements?)? Was inter-annotator agreement computed?
> * All the EMRs are manually annotated by two native speakers (a Master’s degree in computer science and an expert in both the fields of computer science and medicine.). We consider the de-identification task to be easy and straightforward, with very few disagreements during labeling and near-perfect accuracy. Therefore, we did not compute the inter-annotator agreement. However, we employed some post-processing procedure to ensure the accuracy of the annotations. For example, we perform cross-validation on the whole dataset, and manually check the inconsistencies between the predicted PHI mentions and ground-truth.
> * We will perform another round annotation, and report inter-annotator agreement in the revised manuscript. Thanks for your suggestion.
>
>
> > C) What are the descriptive statistics of each data split in the HM corpus?
> * We sincerely appreciate your suggestion. The below table reports the descriptive statistics of HM train/dev/test splits, SY and CCKS datasets. We will replace the current Table 1 with the below table in the revised manuscript.
>
> |   | HM Train | HM Dev. | HM Test | SY | CCKS |
> |  :-- | --: | --:|  --: | --: | --:|
> | **#EMR**  | 300|  100 |100 | 100|  300 |
> | **#Sentence**|  6,005| 2,029 |1,995  | 2,505|  1,346 |
> |**#PHI Mention** |
> | PERSON         |  2,429    | 873      | 841    | 1,923     | 85 |
> |LOCATION       | 220       |  62       | 70       |187       |  219 |
> |HOSPITAL        |  1,091    | 389      | 313     | 233      |  115 |
> |DATE                |  4,755    |  1,458   |1,420    |2,153      | 449 |
> |ID                      | 128       | 32        |31        | 22         |  9 |
> |CONTACT         |  207     |  64        |67       | 36         |  0 |
> |AGE                  |  1,087    | 401      | 369     |253        | 430 |
> |PROFESSION  |  196      | 66        | 66       |15         | 21 |
>
>
> > page 1, line 35: "may be regarded as a easy task" -> "may be regarded as an easy task"
> * Thanks for pointing this out. We will revise it accordingly.
>
>
> 1. Zhang et al. 2020. Conceptualized Representation Learning for Chinese Biomedical Text Mining. *arXiv preprints arXiv*: 2008.10813.
> 2. Aparna et al. 2021. Memorization vs. Generalization: Quantifying Data Leakage in NLP Performance Evaluation. In *EACL 2021*.
> 3. Hendrycks et al. 2020. Pretrained Transformers Improve Out-of-Distribution Robustness. In *ACL 2020*.

---

### Official Review · Reviewer_FbUh · 2023-08-07

**Soundness:** 3

**Excitement:**

3: Ambivalent: It has merits (e.g., it reports state-of-the-art results, the idea is nice), but there are key weaknesses (e.g., it describes incremental work), and it can significantly benefit from another round of revision. However, I won't object to accepting it if my co-reviewers champion it.

**Paper Topic And Main Contributions:**

This paper deals with the EHR de-identification problem in a cross-hospital context. The authors build a corpus (that they say they will make available) with EHRs with three different hospitals and build or finetune different neural models to pursue the de-identification task showing that the cross-hospital results get degraded due to the lack of generalization. They also explore several domain generalization techniques in order to prevent the mentioned problem when there is only the possibility of training with one unique hospital dataset, and they show that only Stochastic Weight Averaging and text smoothing bring slight improvements.

Their main contribution is the corpus they built.

**Reasons To Accept:**

Although the corpus authors built is not available yet, they claim that they will make it available as soon as the paper gets accepted.

**Reasons To Reject:**

The paper does not present any relevant contribution besides the corpus they built. The techniques employed are the standard ones, their conclusion, namely that cross-hospital results are worst than in-hospital results is the expected one. The drop in F1 score is only of a 2% with PLM and increases when adding sequentiality (BiLSTM + CRF) to the system, showing that the structure of the documents probably lies behind that drop, and the generalization techniques proposed do not tackle structural differences.
Finally, as no error analysis is done, a table with the percentage of error of each PHI category might be sufficient, it is not possible to measure to which extend the errors are very relevant because errors not identifying age or the hospital are not the same as error not identifying the names of patients or doctors.

**Reproducibility:**

2: Would be hard pressed to reproduce the results. The contribution depends on data that are simply not available outside the author's institution or consortium; not enough details are provided.

**Reviewer Confidence:**

4: Quite sure. I tried to check the important points carefully. It's unlikely, though conceivable, that I missed something that should affect my ratings.

---

> ### Author Rebuttal · Authors · 2023-08-29
>
> Thank you for your valuable comments. We reply to your concerns as follows:
>
> > The paper does not present any relevant contribution besides the corpus they built. The techniques employed are the standard ones, their conclusion, namely that cross-hospital results are worst than in-hospital results is the expected one.
> * Thanks for your comments. This paper was submitted to the track "Resources and Evaluation"; and indeed, the corpus (and associated evaluation) is our primary contribution. Just like most papers previously published in EMNLP's "Resources and Evaluation" track, new modeling techniques are beyond the scope of our paper.
> * Our corpus has noticeable merits and contributions. As summarized in Lines 58-65, our dataset provides the first multi-source de-identification benchmark. The dataset probably interests researchers from a broader community, since DG tasks have been scarce in NLP. We will publicly release the data.
> * Our evaluation results are also valuable. As you point out, it is an expected finding that "cross-hospital results are worst than in-hospital results", but our evaluation provides *quantitative* instead of *qualitative* results. In practice, the performance drop from 99% to 97% (e.g., HM->SY) is probably acceptable, while drop of 99% to 81% (e.g., HM->CCKS) is probably unacceptable. In that sense, *quantitative* results are more informative and thus important.
>
>
> > The drop in F1 score is only of a 2% with PLM and increases when adding sequentiality (BiLSTM + CRF) to the system, showing that the structure of the documents probably lies behind that drop, and the generalization techniques proposed do not tackle structural differences.
> * We highly appreciate your comments. When a model trained on HM is transferred to SY, the drop in $F_1$ score is 2%; when the model is transferred to CCKS, the drop is about 18%. We agree with you that the document structures are different between datasets. From a neural modeling perspective, such differences cause distribution shifts of neural representations, and finally result in the performance drop.
> * We employ some existing domain generalization (DG) methods, and verify whether they can help the models to generalize across hospitals. Experimental results show that some DG methods can *alleviate*, but *not fully address*, the domain gap issue (Table 3). Hence, our corpus poses a new, realistic, and challenging DG task (Lines 262-265). It highlights the importance of DG research in medical NLP: even a perfectly performing model can encounter dramatic degradation when transferred across hospitals.
> * Negative findings are also implicative in practice. Our results of the ineffectiveness of some DG methods still provide a descent starting point for future research.
>
>
> > Finally, as no error analysis is done, a table with the percentage of error of each PHI category might be sufficient, it is not possible to measure to which extend the errors are very relevant because errors not identifying age or the hospital are not the same as error not identifying the names of patients or doctors.
> * Thank you for the comments. Table 5 (in Appendix C) presents the categorical $F_1$ scores of within- and cross-hospital evaluation. We also provide some analysis on the categorial results (Lines 474-483), which are exactly consistent with your understandings that the performance drop is heterogeneous across PHI categories.

---

### Meta-Review · Area_Chair_XJbG · 2023-09-18

**Recommendation:** 4

**Metareview:**

**Summary:**
This paper investigates the de-identification of clinical notes, with a particular focus on a cross-hospital setting.
The authors have constructed a corpus (which they intend to make available) comprising Electronic Health Records (EHRs) from three different hospitals. They have developed or fine-tuned various neural models to tackle the de-identification task. The authors have evaluated Chinese BERT-based models that have been fine-tuned using this corpus. They have conducted comparisons between same-hospital and cross-hospital settings.
The paper illustrates that, despite achieving high test set performance for "within-hospital" scenarios, there is a significant drop in performance in cross-hospital settings, even for relatively straightforward tasks such as identifying Personal Health Information (PHI) related to specific named entities (e.g., location, patient name, hospital name, patient ID, date/timestamps, patient/physician/hospital contact information, patient age/profession).
In addition, the authors have explored various domain generalization techniques to address this issue when training with a single hospital dataset is the only option. They found that only Stochastic Weight Averaging and text smoothing yield slight improvements.

**Strengths:**
The main contribution of this work lies in the corpus constructed by the authors. While the corpus is not currently accessible, the authors have expressed their intention to release it upon the paper's acceptance.
The dataset holds significant value, given the scarcity of clinical corpora, especially those that encompass cross-hospital and non-English contexts. Furthermore, this work merits recognition for its role in highlighting the generalization gap observed in deep learning models.

**Weaknesses:**
Reviewers have identified the following weaknesses in the paper:
1. The techniques employed by the authors are standard. It is challenging to assess the significance of errors since errors in identifying age or the hospital are not equivalent to errors in identifying patient or doctor names.
2. The authors opted to create their medical BERT-based model (HM-BERT) while disregarding existing pre-trained medical models. It would have been valuable to explore alternative models for comparison.
3. The paper lacks an in-depth analysis of the aspects contributing to the generalization gap. For instance, there is insufficient exploration of the similarities and differences between the data from different hospitals that might explain the drop in model performance. It is possible that the document structure plays a role in this drop, and the proposed generalization techniques do not address structural differences.
4. The paper's conclusion, stating that cross-hospital results are worse than in-hospital results, is expected and does not provide a novel insight.

**Author-Reviewer discussion and acknowledgment:**
During the rebuttal phase, the authors have responded to the questions and concerns raised by the reviewers by providing clarifications and outlining their planned improvements to the paper.

**Conclusion:**
The paper is well-written and clear, offering valuable insights to the community, particularly in the context of medical NLP applications. However, reviewers recommend that the authors incorporate additional references and enhance the paper based on the points raised during the discussion phase. Additionally, the authors should address the identified typos.

---

### Decision · Program_Chairs · 2023-10-07

**Decision:**

Accept-Main

**Comment:**

**Summary:**
This paper investigates the de-identification of clinical notes, with a particular focus on a cross-hospital setting.
The authors have constructed a corpus (which they intend to make available) comprising Electronic Health Records (EHRs) from three different hospitals. They have developed or fine-tuned various neural models to tackle the de-identification task. The authors have evaluated Chinese BERT-based models that have been fine-tuned using this corpus. They have conducted comparisons between same-hospital and cross-hospital settings.
The paper illustrates that, despite achieving high test set performance for "within-hospital" scenarios, there is a significant drop in performance in cross-hospital settings, even for relatively straightforward tasks such as identifying Personal Health Information (PHI) related to specific named entities (e.g., location, patient name, hospital name, patient ID, date/timestamps, patient/physician/hospital contact information, patient age/profession).
In addition, the authors have explored various domain generalization techniques to address this issue when training with a single hospital dataset is the only option. They found that only Stochastic Weight Averaging and text smoothing yield slight improvements.

**Strengths:**
The main contribution of this work lies in the corpus constructed by the authors. While the corpus is not currently accessible, the authors have expressed their intention to release it upon the paper's acceptance.
The dataset holds significant value, given the scarcity of clinical corpora, especially those that encompass cross-hospital and non-English contexts. Furthermore, this work merits recognition for its role in highlighting the generalization gap observed in deep learning models.

**Weaknesses:**
Reviewers have identified the following weaknesses in the paper:
1. The techniques employed by the authors are standard. It is challenging to assess the significance of errors since errors in identifying age or the hospital are not equivalent to errors in identifying patient or doctor names.
2. The authors opted to create their medical BERT-based model (HM-BERT) while disregarding existing pre-trained medical models. It would have been valuable to explore alternative models for comparison.
3. The paper lacks an in-depth analysis of the aspects contributing to the generalization gap. For instance, there is insufficient exploration of the similarities and differences between the data from different hospitals that might explain the drop in model performance. It is possible that the document structure plays a role in this drop, and the proposed generalization techniques do not address structural differences.
4. The paper's conclusion, stating that cross-hospital results are worse than in-hospital results, is expected and does not provide a novel insight.

**Author-Reviewer discussion and acknowledgment:**
During the rebuttal phase, the authors have responded to the questions and concerns raised by the reviewers by providing clarifications and outlining their planned improvements to the paper.

**Conclusion:**
The paper is well-written and clear, offering valuable insights to the community, particularly in the context of medical NLP applications. However, reviewers recommend that the authors incorporate additional references and enhance the paper based on the points raised during the discussion phase. Additionally, the authors should address the identified typos.